# Graph-Based Prediction of Meeting Participation

**Gabriel Murray** 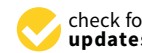

Department of Computer Information Systems, University of the Fraser Valley, Abbotsford, BC V2S 7M8, Canada; gabriel.murray@ufv.ca

**Abstract:** Given a meeting participant's turn-taking dynamics during one segment of a meeting, and their contribution to the group discussion up to that point, our aim is to automatically predict their activity level at a later point of the meeting. The predictive models use verbal and nonverbal features derived from social network representations of each small group interaction. The best automatic prediction models consistently outperform two baseline models at multiple time-lags. We analyze which interaction features are most predictive of later meeting activity levels, and investigate the efficacy of the verbal vs. nonverbal feature classes for this prediction task. At long time-lags, linguistic features become more crucial, but performance degrades compared with prediction at short time-lags.

**Keywords:** group interaction; social network analysis; social signal processing; natural language processing; conversational data; multimodal interaction

## 1. Introduction

The core question in this research is whether we can automatically predict a conversation participant's activity levels at a later point in the conversation, given features that describe their current turn-taking dynamics and contributions to the discussion so far. Such predictions could be useful for an automated meeting assistant that provides feedback on interaction patterns, with the goal of encouraging participation. A system with this capability could be used in real-time during a meeting discussion, or as an offline tool to assist a team leader or manager in analyzing which factors lead to greater participation by various team members.

The machine learning models used for this task utilize both verbal and nonverbal features of the discussion. Both classes of features are derived from graph-based representations of the conversation, and the features include measures of network centrality and network change. The nonverbal features represent the turn-taking dynamics during a particular slice of the conversation, based on a graph representation of turn transitions. The verbal features capture aspects of topic structure, linguistic alignment, and lexical change during the conversation, based on a tripartite graph representation linking speakers, words, and sentences.

The prediction task is carried out at multiple time-lags, where a time-lag is how far into the future we are making a prediction. The best-performing machine learning models are significantly better than two baselines at all time-lags, though the task is unsurprisingly more difficult at longer time-lags. At short time-lags, simple nonverbal features such as the current activity level of the target participant are very effective on their own, while at longer time-lags the verbal/linguistic features become more important for predictive accuracy. We provide an in-depth analysis of the most effective features and machine learning models.

*Related Work*

Recent related work includes a similar time-lagged prediction task by Stewart et al. [1], focused on time-lagged prediction of speech rate by participants in problem-solving groups. In contrast, we

are predicting a participant's activity level at a later point in the meeting, with their activity level measured as the number of sentences they speak during that later slice of the meeting conversation. Murray [2] also carried out experiments on time-lagged prediction of a participant's activity level, but used only nonverbal features related to turn-taking. In contrast, we use nonverbal turn-taking features as well as linguistic features relating to topic structure and linguistic alignment.

Related work on predicting activity in a discussion includes predicting who the next speaker will be [3] and predicting when a turn will end [4]. Kim and Rudin [5] apply machine learning to meetings in order to answer several research questions, including prediction of how long a meeting will continue once the key decisions have been made. There has been work on using social network analysis and graph-based methods for modelling turn-taking, modelling influence, and detecting roles or traits in social interaction [6–8], but to our knowledge such models have not been used for predicting future activity levels in a conversation, as is proposed in this work.

There has been relevant work on detecting involvement and "hot spots" in meetings, with a hot spot being variously defined as a short period of time during which multiple participants are highly active [9], or a portion of the meeting containing many summary-worthy dialogue acts (sentence-like units) [10]. In contrast, our work focuses on predicting the activity levels of individual participants. However, aspects of group involvement are captured by some of the features used in this task.

Our work has much in common with work in social signal processing (SSP) [11], in that we are applying machine learning to a prediction task in the domain of small group interaction and we utilize nonverbal features. However, most SSP work exclusively focuses on nonverbal aspects of interaction, whereas we include linguistic features as well and want to determine the circumstances under which these features are useful for the task at hand. Examples of SSP research on conversational data include automatic prediction of group performance on a task, using nonverbal features of the conversation [12], or a combination of verbal and nonverbal features [13]. Lai and Murray [14] also combine verbal and nonverbal features to predict aspects of group satisfaction and affect. Modelling emergent leadership in meetings has become a major research topic in SSP as well [15,16], with most approaches focusing on nonverbal aspects of the group interaction.

The current work has some relation to work on linguistic alignment [17,18] or entrainment [19,20], wherein conversation participants' language becomes more similar over the course of an interaction. Our graph-based approach to language embeds the participants in a graph that also includes sentences and words, and this representation allows us to ask questions such as how similar two participants are linguistically, whether they exist in the same cluster (community), and how their language network is changing as the conversation progresses. However, linguistic alignment is not the primary focus of the current paper.

The structure of the remainder of the paper is as follows. In Section 2, we describe the graph-based models used to represent the conversational turn-taking and discussion. In Section 3 we describe the corpus and experimental setup. We present the experimental findings in Section 4, and conclude in Section 5.

## 2. Social Network Models

In this section we describe the two types of graph-based social network models used in this research. The first class of models is used to represent conversational turn-taking, and these do not incorporate linguistic features. The second class of models explicitly represents the language being used by conversation participants. Features are derived from both models and combined to form a multimodal perspective on the conversational interaction.

In these experiments we use the AMI meeting corpus [21], where each meeting consists of four people role-playing as a team designing a product (a remote control). Each group goes through a series of four meetings. Each participant is assigned a role, as a project manager, industrial designer, user interface designer, or marketing expert.

All of the graph models begin by segmenting each meeting into non-overlapping windows of 20 dialogue act units each. We utilize the manual dialogue act segmentations released as part of the AMI corpus. Choosing 20 dialogue act units as the window-size is motivated by the fact that it is large enough to likely feature multiple participants within each window, and small enough that each window will foreground a small segment of a much larger conversation. We take turns treating each participant as the target participant whose later participation levels we want to predict. For each window, features are extracted relating to the target participant and other participants, as described in the subsections below. We then predict the sentence frequency (or *dialogue act* frequency) of the target participant after a time lag of $n$ windows, where we vary $n$ with the values 1, 2, 3, 4, 5. That later window is the *target window*. For example, a time-lag of 1 means we are predicting the target participant's dialogue act frequency in the subsequent window of the conversation. A time-lag of 5 means we are are predicting their dialogue act frequency after 80 ($4 \times 20$) intervening dialogue acts in the conversation.

### 2.1. Turn-Taking Graphs

Within each window of 20 dialogue act units, we represent the group interaction dynamics in that window as a directed graph, with nodes representing participants. There is an edge $(A, B)$ between participants $A$ and $B$ if there is at least one immediate transition within that window from $A$'s speaking turn to $B$'s speaking turn. The edge $(A, B)$ has a cost which is the reciprocal of the number of times that transition was made within the current window.

After building the graph for the current window, we extract betweenness centrality, closeness centrality, and degree centrality features for each participant. Specifically, the features (and abbreviations, for later reference) are as follows:

- The betweenness centrality, degree centrality, and closeness centrality of the target speaker in the current window (*bet_targspeaker*, *deg_targspeaker*, and *close_targspeaker*, respectively).
- The minimum, maximum and mean betweenness centrality of the other (non-target) participants in the current window (*bet_oth_min*, *bet_oth_max*, and *bet_oth_mean*, respectively).
- The minimum, maximum and mean degree centrality of the other (non-target) participants in the current window (*deg_oth_min*, *deg_oth_max*, and *deg_oth_mean*, respectively).
- The minimum, maximum and mean closeness centrality of the other (non-target) participants in the current window (*close_oth_min*, *close_oth_max*, and *close_oth_mean*, respectively).

These features are extracted because we suppose that a participant's centrality in the turn-taking graph will reflect aspects of their assigned role, their emergent leadership [15,16], and their conversational dominance, and that these attributes should be predictive of their longer-term behaviour in the meeting. For example, somebody who is emerging as a leader and taking turns addressing each of their teammates might tend to continue being active and engaged later in the meeting.

### 2.2. Language Graphs

For each window, a language graph is built that represents the group conversation up to and including the current window. Conceptually, this is a tripartite hypergraph, where each edge connects three nodes representing three types of entities: word *types* (all occurrences of a particular word belong to the same *type*), dialogue act units, and speakers. A given edge therefore represents a word used in a particular dialogue act unit by a particular speaker (i.e., it represents a word *token*).

The tripartite hypergraph graph as just described can also be represented by an equivalent bipartite graph. In this work, we use the bipartite representation to extract the features described below. Each tripartite hypergraph $H$ is transformed into a bipartite graph $B$ as follows. The bipartite graph $B$ has partitions $N$ and $E$, where the nodes of $N$ are the nodes from the original hypergraph $H$, and the nodes of $E$ are the edges of the original graph $H$. The resulting graph $B$ will contain an edge $< n_1, e_1 >$ connecting nodes of $N$ and $E$ if $n_1$ belonged to edge $e_1$ in the original hypergraph $H$.

Figure 1 shows an example bipartite language graph generated partway through one of the meeting discussions used in these experiments. The nodes representing the four participants are highlighted, as are the edges connected to those four nodes. The nodes in the left partition that are not highlighted correspond to the word types nodes and dialogue act nodes from the original graph *H*, while the nodes in the right partition correspond to edges in *H*.

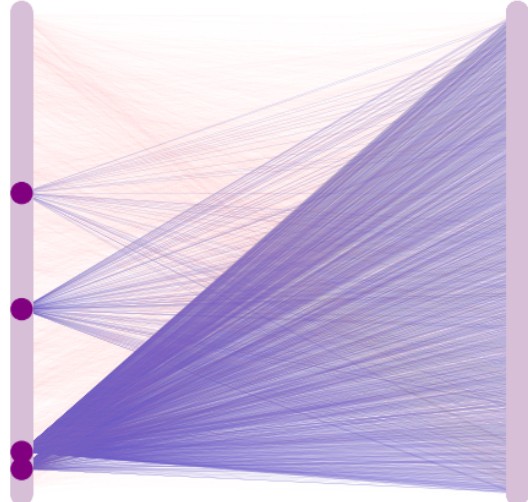

**Figure 1.** Bipartite Language Graph *B* with Participant Nodes/Edges Highlighted. *Left Side:* Nodes in *B* that were also nodes in the original hypergraph *H*. *Right Side:* Nodes in *B* that were edges in the original hypergraph *H*.

One benefit of this graph transformation is that, in the bipartite representation, the word tokens are now represented as nodes (e.g., the right side of Figure 1), rather than edges. This has an impact on some of the extracted features described below, such as those relating to the community structure of the graph, as communities will now consist of speakers, dialogue act segments, word types, and word tokens. Word tokens of the same type can belong to different communities, giving us a graph representation that is better able to capture distinctions such as lexical polysemy (e.g., two occurrences of the word "bank" with different meanings) or differing word usage patterns by different participants.

Many of the features we extract from the language graph involve the betweenness centrality of each participant in the language graph. However, this centrality measure will tend simply to correlate with the frequency of the participant's dialogue acts in the conversation up to that point. For example, the participant represented by the dark node in the upper-left of Figure 1 has participated less often in the conversation so far, and will likely have a lower betweenness centrality as a result. It is possible for a group member to have high participation levels but low centrality, e.g., if they tend to discuss different topics than the others, or have a different or more limited vocabulary, but in general participation frequency and language graph centrality will correlate. For that reason, we divide the centrality measure of each participant by their participation frequency, i.e., by the proportion of dialogue act units in the conversation so far that belong to them.

We extract the following language graph centrality features:

- The linguistic centrality of the target speaker (*lingcent_targspeaker*), rescaled by participant frequency.
- The minimum, maximum, and mean linguistic centrality scores of the other (non-target) participants (*lingcent_oth_min*, *lingcent_oth_max*, and *lingcent_oth_mean*, respectively), again rescaled based on participant frequency.

These features are extracted because participants who have high centrality in the language graph (even after normalizing for their participation frequency) may have linguistic alignment with the other participants, and are more likely to have contributed to many of the topics that have been discussed up to that point. We hypothesize that these attributes will have a bearing on how much they will participate at later points in the meetings.

Other features that are extracted involve community detection (or clustering) on the language graph. We use modularity-based community detection [22], where each node begins in a singleton community, and each step consists of merging two communities that most increase the modularity, until the modularity cannot be improved. One advantage of this approach is that the number of communities does not need to be pre-specified, and can be used as a predictive feature. Because the original graph is a tripartite hypergraph, each of the resulting communities can contain a mix of participant nodes, sentence nodes, and word nodes (types and tokens). A qualitative analysis of these communities shows that there is typically a small number $n$ (where $n \approx$ the number of participants) of large communities where each of these large communities roughly corresponds to the vocabulary of a particular participant. There is then a larger number of small communities, each of which roughly corresponds to a topic in the discussion. Recent research [23] shows the advantages of taking a graph-based approach to topic detection and also describes potential weaknesses of popular alternative approaches such as Latent Dirichlet Allocation (LDA) [24]. However, our current work differs from that recent research in that our graphs include not only words and sentences but participants as well, and so the resulting community structure relates not only to topics but also to lexical similarities and differences between the conversation participants.

The features that relate to community structure are as follows:

- The total number of communities (*num_comms*).
- The number of communities represented by sentences in the current window (*num_win_comms*). This will range from 1 (all sentences in the current window belong to one community) to 20 (each sentence in the window is in a different community).
- The number of communities containing speakers (*num_speak_comms*). For this corpus, this feature will range between 1 (all speakers in the same community) to 4 (each speaker in a separate community).

We also extract the following feature relating to the changing language network:

- The change in density of the language network between the previous and current windows (*density_change*). Density is a measure of how connected the graph is, e.g., a graph with the maximum possible number of edges has a density of 1.

*2.3. Other Features*

Finally, we extract the remaining features that are not derived from either of the above graph-based models.

- The frequency (number of dialogue act units) of the target speaker in the current window (*freq_targspeaker*).
- The minimum, maximum and mean frequency of the other (non-target) participants in the current window (*freq_oth_min*, *freq_oth_max*, and *freq_oth_mean*, respectively).
- The feature *loc* is the current index/location in the conversation.

## 3. Experimental Setup

We partition the AMI corpus so that there are 86 meetings in the training set and 52 in the test set. We ensure that each group has its meetings contained entirely in the training set or entirely in the test set, to make the prediction task more challenging. The number of actual observations in

the training and test set depends on the value of the time lag $n$. For example, with $n = 1$, there are 12,872/7696 training/test examples, and with $n = 5$ there are 11,496/6864 training/test examples.

In these experiments, we use linear regression, random forest, and gradient boosting models. These models are chosen as they tend to perform well on small-to-medium amounts of data. All the machine learning models are built using sci-kit learn (https://scikit-learn.org/stable/index.html). For both random forest and gradient boosting, the number of estimators was set at 30. Otherwise, we used the default parameters in sci-kit learn and did not perform parameter optimization.

These models are contrasted with a first baseline that predicts the mean dialogue act frequency from the training data, and a second baseline that predicts that the target dialogue act frequency will be the same as the current dialogue act frequency. This latter baseline assumes that a participant's activity level will stay roughly the same between the current window and target window. For evaluation, we use mean-squared error (MSE) and $R^2$ goodness-of-fit scores.

## 4. Results

Figure 2 shows the MSE values for all models and all time-lags, when using all of the extracted features. The linear regression model and gradient boosting model are the best overall prediction models, and perform very comparably to each other. They are significantly better than both baselines across all time lags (all $p < 0.05$). However, the improvement over both baselines at lag 1 is very substantial (e.g., a 36% relative reduction in MSE over the mean prediction baseline), while the improvement over the mean prediction baseline at lag 5 is very small. Generally, all of the prediction methods perform worse as the time-lag between the current window and the target window gets larger, with the exception of the mean prediction baseline. The baseline that predicts the participant's participation level will stay the same in the target window as the current window gets much worse as the time-lag grows.

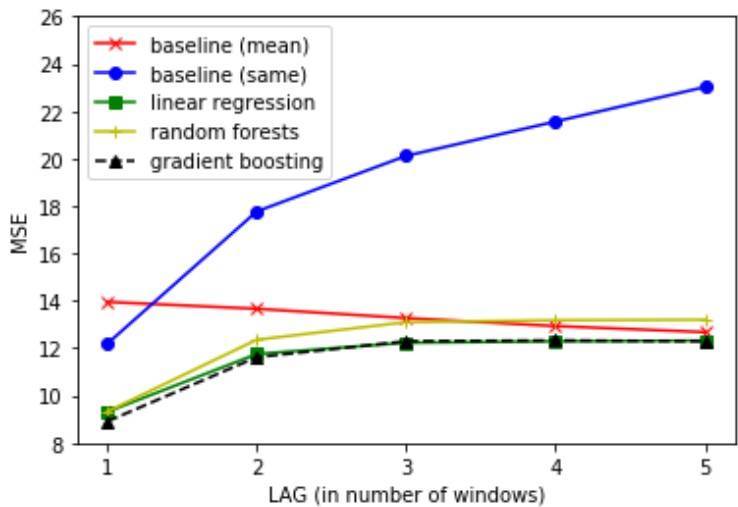

**Figure 2.** Mean-squared error (MSE) for all lags using all features.

Figure 3 shows the top 10 most important features at time-lag 1 and time-lag 5. Feature importance is determined by the average reduction in MSE when the feature is used as a split-point in a decision tree in the gradient boosting model. At time-lag 1, the most predictive features by far are simply the frequency of the target participant and the average frequency of the other participants. The number of communities in the language graph is the next most important feature, and all of the other features have much lower importance scores. In contrast, at time-lag 5 the graph-based features are much more important. In particular, the linguistic centrality scores of the target participant and other participants all have much higher importance. The density change in the language graph is also important, as are the betweenness measures from the turn-taking graph.

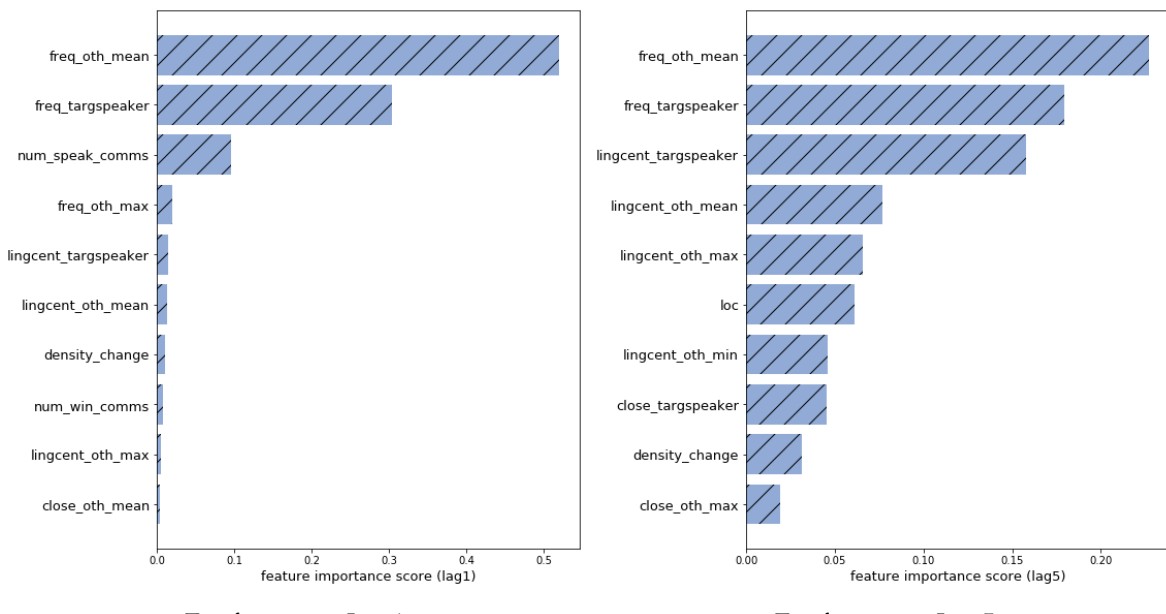

Top features at Lag 1                                          Top features at Lag 5

**Figure 3.** Top features at different time lags.

As the linguistic centrality of the target speaker (*lingcent_targspeaker*) is highly important for predicting the outcome variable at lag 5, we analyze the correlation between these variables in the training data. Figure 4 shows a scatterplot of the two variables, with the current linguistic centrality of the target speaker on the x-axis and their later activity (at lag 5) on the y-axis. There is a very weak but highly significant positive correlation between the two variables (Spearman's rho = 0.04, $p < 0.001$). That is, a participant who has high language graph centrality, even after rescaling to account for their participation frequency, will have a slight tendency, on average, to be more active at the later point of the meeting. One can also see outliers in the lower-right of the graph, where some participants have high language graph centrality at one point of a conversation but are not active later in the meeting during the target window.

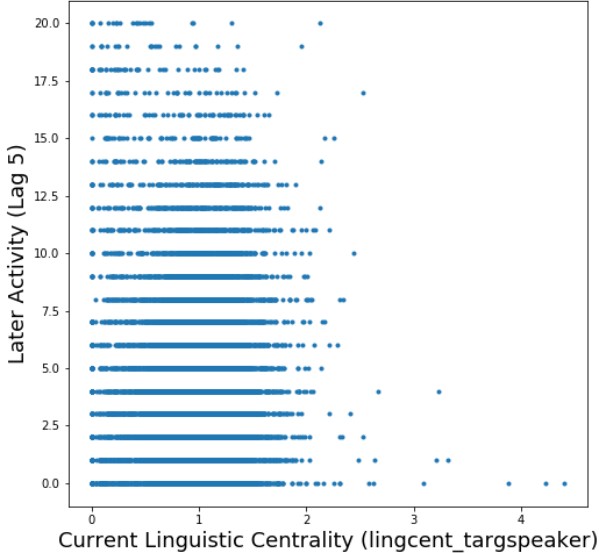

**Figure 4.** Normalized Linguistic Centrality vs. Activity at Lag 5.

We next turn to analyzing the usefulness of individual feature subclasses. Figure 5 shows the MSE results for all models and all time-lags when only using the features extracted from the turn-taking graphs and when only using features extracted from the language graph. In both cases, the MSE values are higher than when using the full feature set, and are closer to the performance of the baseline models. When each feature class is used on its own, the turn-taking features are slightly more useful than the language graph features. However, both are outperformed by using the multimodal models that use all of the features.

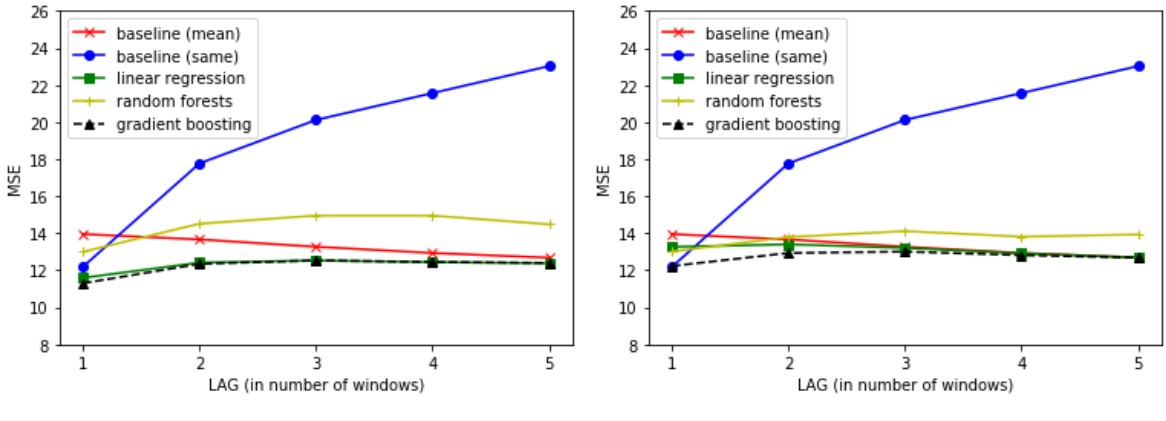

MSE for all lags using turn-taking features        MSE for all lags using linguistic features

**Figure 5.** Comparing feature subsets.

Table 1 shows the $R^2$ scores for the gradient boosting models at all time-lags, when using all of the features. At a time-lag of 1, the $R^2$ score is 0.35, meaning that the trained model explains about 35% of the variation in the data. However, the $R^2$ values steadily decrease as the time-lag increases, again illustrating that the prediction task is very challenging over long time-lags. This provides motivation for investigating additional features that can be brought to bear on this task.

**Table 1.** $R^2$ Values for the best models.

| Lag | $R^2$ |
|-----|-------|
| 1 | 0.35 |
| 2 | 0.17 |
| 3 | 0.11 |
| 4 | 0.08 |
| 5 | 0.08 |

One limitation of the current work is that it is applied to a dataset where the participants have assigned roles, and experimental results may differ for groups with no assigned or well-defined roles. However, in the experiments described in this paper, the predictive features did not use any information about the assigned roles of the participants. Furthermore, a participant in the AMI corpus can still emerge as a leader (e.g., by taking an active and guiding role) even if they are not assigned the role of project manager. Future work will apply these methods to other datasets featuring task-based group interaction where roles are not assigned, e.g., the Emergent Leadership (ELEA) corpus [15] and the Group Affect and Performance (GAP) corpus [25]. We hypothesize that the group interaction patterns will differ in these corpora compared to the AMI corpus, as roles (such as *leader*) can emerge more naturally. These differing patterns may be reflected in the turn-taking and language graphs that we use to represent the group conversation.

A second limitation is that the AMI corpus uses an artificial scenario for the participants (as do the ELEA and GAP corpora). While the speech is spontaneous rather than scripted, and groups

have flexibility in their decision-making, there is a set structure in terms of the steps that each group has to go through. We assume that this also has an impact on the manner in which group members interact with one another. In future work, we will apply these modelling and prediction methods to naturalistic, non-scenario group interactions as well, e.g., the D64 corpus [26].

## 5. Conclusions

In these experiments, we have addressed the task of automatically predicting a group member's participation level at a later point in a meeting, given features representing their current turn-taking dynamics and their contribution to the conversation so far. Most of the features are derived from graph representations of turn-taking and of the language used in the conversation. The best automatic prediction models consistently outperform two baseline models at multiple time lags, though the task is very challenging at long-time lags and the models approach baseline performance. At those long time-lags, the linguistic features become more important to overall performance. The models using all features consistently outperform the models that only use turn-taking features or only use linguistic features.

The difficulty of the prediction task at long time-lags provides motivation for finding additional features that can be brought to bear on this task. Many additional features can be derived from the existing turn-taking and language graphs. For example, in future work we will explore graph-based similarity measures for comparing participant nodes in the language graph. We will also carry out research on how best to interpret and visualize the discovered communities in the language graph. Finally, we will investigate how to incorporate these graph-based models and their predictions into an automated meeting assistant that can explain group interaction patterns. The goal is to develop a system that can explain the factors that encourage and discourage participation in a meeting.

**Acknowledgments:** The author is supported by an NSERC Discovery Grant [RGPIN-2018-06806].

**Conflicts of Interest:** The author declares no conflicts of interest.

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
