# Peer review of "Graph-Based Prediction of Meeting Participation"

_mti, doi:10.3390/mti3030054_

Round 1

Reviewer 1 Report

This paper answers the question of whether participant activity levels later in the conversation can be predicted from current turn-taking dynamics and contributions to the discussion. The authors used the AMI meeting corpus to detect sentence frequency of a speaker at a later (lagged) time window. To do this, they created graphs of the turn taking dynamics of the conversation, and calculated measures of the graph’s structure (e.g. betweenness centrality, degree centrality). They compared MSE of regression, random forest, and gradient boosting models to a baseline that predicted the mean frequency, and another that predicted the current speaker’s frequency. They found that their best performing models outperformed both baselines at all lags, with shorter time lags yielding greater accuracy.

The work in this paper clearly demonstrates that the fusion machine-learning and graph-based approach can feasibly model activity levels in group conversation. The baselines were clear and reasonable, and convincingly demonstrated performance over baselines at shorter lags.

This paper could be improved in four key ways. First, the related work section is lacking. The structure is a bit hard to follow and key research areas are left out, like next-speaker prediction.

Second, the choice of features to use in the model seems arbitrary, specifically the centrality and community structure measures. There is no explanation about why these features were theoretically chosen and how each one provides unique insights into the turn-taking dynamics.

Third, some explanations need improving. Specifically, the explanation of how the language graphs were formed is hard to follow. The introduction of the corpus comes too late in the paper. The description of the input features does not make as much sense without knowing what the dataset entails. There are no details on the machine learning models and parameters themselves.

Finally, the paper is lacking in explanation as to why certain features were predictive. The paper would be stronger by adding intuition about why these features provide insight into the participation dynamics. Additionally, a comparison of the individual modalities would strengthen claims of which features are predictive. The best performing model could be used on each feature set (i.e. centrality measures versus community structure measures), and prediction accuracy discussed.  

Author Response

Thank you kindly for the valuable feedback. The paper has benefited greatly from your comments. Please see the attached responses to your review. 

Reviewer 2 Report

-       Section 1 and 2 should be merged according to the manuscript guideline for this journal.  

-       Related work should be extended and more references should be used. The limitations of the previous approaches should also be discussed in more details.

-       In section 3 (Social network models), clarification is needed in the second paragraph. Furthermore, why do you think 20 dialogue act units is a good number?

-       Figure 1 is low quality. It should be regenerated and axis titles should be added.

-       In section 4, more clarification is needed about why linear regression, random forest and gradient boosting models have been chosen.

-       Limitations of the approach should be described.

-       In general, a paper with 8 pages and 14 refrences is a little short for a journal and it is more suitable for a conference proceedings.

Author Response

Thank you very much for your review. The paper has greatly benefited from your comments. Please see the attached responses to your review. 

Round 2

Reviewer 1 Report

The related works has improved, quite a bit and now discusses social signal processing, next speaker prediction, and other related research. The only missing discussion is whether or not other work has taken a graph-based approach to similar topics (next speaker prediction, turn taking dynamics, etc).

It is still unclear to me where the dialogue acts come from? Are they human codes or automatically generated? Please explain the ground truth further.

The explanation of the graph construction methodology has improved. However, the following questions still remain.

1) What are the word types in the tripartite graph? Are they dictionary categories or part of speech?

2) Why was the tripartite graph turned into a bipartite graph. Couldn’t the same features still be extracted from the tripartite graph?

Were any other machine learning parameters tried when training the classifiers?

Finally, please discuss your findings in terms of a future application for the models. It is unclear what the contribution is above and beyond similar works. How can these models uniquely be applied to dialogues?

Author Response

Thank you to the reviewer for your feedback. Please see the attached document containing responses that address each point. 

Round 3

Reviewer 1 Report

This paper describes a graph-based approach to detecting activity level in meetings. A number of language and dialogue features were computed and used as input to linear regression, random forest, and gradient boosting classifiers. The authors found that all models outperformed chance, but accuracy was predictably better for targets closest to the input data. Further, multimodal models performed better than unimodal models.

Overall this paper has greatly improved in style and content. I have a few minor comments and one more major comment for the authors to address.

Major Comments:

Overall, the paper is lacking a substantial discussion. There is little explanation of how these models could be implemented in real-time systems, and how they would impact our current understanding of meeting dynamics. There is a lack of discussion relating the findings back to theory, which would greatly strengthen the paper.

Minor Comments:

The author says “The edge (A, B) has a cost which is the reciprocal of the number of times that transition was made within the current window.: Why did you weight the graph like this? It seems as though more common transitions should have higher weights. However, this depends on what phenomena you are trying to encode with the graph, which is not fully explained.

The author says “The tripartite hypergraph graph as just described can also be represented by an equivalent bipartite graph.”

I believe the second “graph” is a typo.

What is the sample size in Figure 4? It looks as if there are multiple observations per participant. This will violate the assumption that each sample is independent, and probably inflate your p value. The correlation should be done with one value per participant. Perhaps create an overall centrality score and activity score per participant and correlate those.